Transfemoral limb loss modestly increases the metabolic cost of optimal control simulations of walking

Miller Ross H. rosshm@umd.edu 1 2
Bell Elizabeth M. 1 3
Russell Esposito Elizabeth 4 5 6 7 8
1 Department of Kinesiology, University of Maryland at College Park , College Park , MD , United States of America
2 Neuroscience and Cognitive Science Program, University of Maryland , College Park , MD , United States of America
3 Department of Kinesiology, Towson University , Towson , MD , United States of America
4 Military Operational Medicine Research Program , Fort Detrick , MD , United States of America
5 Extremity Trauma and Amputation Center of Excellence , Fort Sam Houston , TX , United States of America
6 Center for Limb Loss and Mobility, VA Puget Sound Healthcare System , Seattle , WA , United States of America
7 Madigan Army Medical Center , Tacoma , WA , United States of America
8 Department of Physical Medicine and Rehabilitation, Uniformed Services University of Health Sciences , Bethesda , MD , United States of America
Hutchinson John
Electronic publication date: 2024 Jan 9
Publication date: 2024
Volume: 12
Electronic Location ID: e16756
Received 2023 Aug 22; Accepted 2023 Dec 13
Copyright: ©2024 Miller et al.
Copyright year: 2024
Copyright holder: Miller et al.
License: This is an open access article, free of all copyright, made available under the Creative Commons Public Domain Dedication. This work may be freely reproduced, distributed, transmitted, modified, built upon, or otherwise used by anyone for any lawful purpose.
License URL: https://creativecommons.org/publicdomain/zero/1.0/

Keywords: Amputation, Mobility, Prosthesis, Above-knee, Energy expenditure, Gait, Direct collocation, Optimization, Muscle strength, Deviations

Funding: Telemedicine & Advanced Technology Research Center DoD-VA Extremity Trauma and Amputation Center of Excellence (EACE) This study was supported by funding from the Telemedicine & Advanced Technology Research Center’s (TATRC) Advanced Medical Technology Initiative (AMTI) award (Bethesda, MD) and by the DoD-VA Extremity Trauma and Amputation Center of Excellence (EACE). The funders had no role in study design, data collection and analysis, decision to publish, or preparation of the manuscript.

==============================
Background

In transtibial limb loss, computer simulations suggest that the maintenance of muscle strength between pre- and post-limb loss can maintain the pre-limb loss metabolic cost. These results are consistent with comparable costs found experimentally in select cases of high functioning military service members with transtibial limb loss. It is unlikely that similar results would be found with transfemoral limb loss, although the theoretical limits are not known. Here we performed optimal control simulations of walking with and without an above-knee prosthesis to determine if transfemoral limb loss per se increases the metabolic cost of walking.

Methods

OpenSim Moco was used to generate optimal control simulations of walking in 15 virtual “subjects” that minimized the weighted sum of (i) deviations from average able-bodied gait mechanics and (ii) the gross metabolic cost of walking, pre-limb loss in models with two intact biological limbs, and post-limb loss with one of the limbs replaced by a prosthetic knee and foot. No other changes were made to the model. Metabolic cost was compared between pre- and post-limb loss simulations in paired t-tests.

Results

Metabolic cost post-limb loss increased by 0.7–9.3% (p < 0.01) depending on whether cost was scaled by total body mass or biological body mass and on whether the prosthetic knee was passive or non-passive.

Conclusions

Given that the post-limb loss model had numerous features that predisposed it to low metabolic cost, these results suggest transfemoral limb loss per se increases the metabolic cost of walking. However, the large differences above able-bodied peers of ∼20–45% in most gait analysis experiments may be avoidable, even when minimizing deviations from able-bodied gait mechanics. Portions of this text were previously published as part of a preprint (https://www.biorxiv.org/content/10.1101/2023.06.26.546515v2.full.pdf).

Introduction

Loss of a lower limb above the knee, i.e., transfemoral limb loss, is associated with a variety of deviations from able-bodied mechanics and energetics of walking. Examples include hip-hiking and a relatively high metabolic cost (Carse et al., 2020). Gait deviations are thought to play a role in mobility, quality of life, and risk for secondary physical conditions in individuals with limb loss (Gailey et al., 2008). High energy expenditure of walking in older adults has been associated prospectively with slowing of walking speed (Schrack et al., 2016), hippocampus deterioration (Dougherty et al., 2022), and increasing perceived fatiguability (Schrack et al., 2020). Avoiding gait deviations and maintaining an economical gait are therefore important clinical goals following limb loss.

Military service members with transtibial limb loss do not have high metabolic costs of walking compared to age-matched able-bodied service members walking at similar speeds (Russell Esposito et al., 2014; Jarvis et al., 2017). The ability of these individuals to avoid high metabolic costs was speculatively attributed to their strength and fitness relative to the general limb loss population. This speculation was supported by optimal control simulations, where replacing one of a musculoskeletal model’s lower limbs with a transtibial prosthesis did not increase the metabolic cost of walking with minimal gait deviations if the strength of the remaining muscles was retained (Russell Esposito & Miller, 2018; Miller & Russell Esposito, 2021). Therefore, transtibial limb loss per se did not increase the metabolic cost of walking. The assumption of no change in muscle strength following limb loss may not always be realistic or feasible, but it allows for isolation of the effect of the non-modifiable factor (loss of a limb) from modifiable factors (e.g., strength), within the limitations of the modeling assumptions. However, these findings are unique to transtibial limb loss. Service members with transfemoral limb loss still had a higher metabolic cost than age-matched able-bodied controls (Jarvis et al., 2017; Russell Esposito, Ràbago & Wilken, 2018). Individuals with transtibial vs. transfemoral limb loss have different relationships between metabolic cost and walking mechanics, such as the cost of lateral balance (Ijmker et al., 2014) and the cost of increasing speed (Genin et al., 2008), suggesting it is more challenging for individuals with transfemoral limb loss to maintain a low metabolic cost without large gait deviations. It is therefore unclear if the finding that limb loss per se does not increase the metabolic cost of walking also pertains to transfemoral limb loss.

Therefore, the purpose of this study was to perform optimal control simulations of walking to isolate the effect of unilateral transfemoral limb loss on metabolic cost. Based on previous experiments on military service members (Jarvis et al., 2017; Russell Esposito, Ràbago & Wilken, 2018) and similar findings on other young, physically fit individuals with transfemoral limb loss (Gitter, Czerniecki & Weaver, 1995), we expected that replacing one of a three-dimensional musculoskeletal model’s limbs with a transfemoral prosthetic limb would increase the metabolic cost of walking when the model attempted to minimize a weighted summation of metabolic cost and deviations from able-bodied walking mechanics.

Materials & Methods

Pre-limb loss model description

Simulations of walking were performed using a three-dimensional model of the human musculoskeletal system with 31 degrees of freedom and 120 Hill-based muscle models (Fig. 1). The model was a modified version of the model described by Rajagopal et al. (2016), implemented in version 4.4 of OpenSim software (Delp et al., 2007). The following modifications were made to the original model to accommodate use in gradient-based optimal control simulations with OpenSim Moco software (Dembia et al., 2020):

Figure 1 Visualization of OpenSim model.

(A) Pre-limb loss version of the OpenSim model; (B) the model modified to include transfemoral limb loss at the right thigh; (C) plantar surface of the model’s foot with contact spheres.

1. The muscles were changed to “DeGrooteFregly2016”-type muscles, which have dynamics suitable for gradient-based optimization (De Groote et al., 2016).

2. Muscle parameters were adjusted as described by Lai, Arnold & Wakeling (2017) to include more realistic knee muscle moment arms at up to 140°  of knee flexion. Although 140°  of knee flexion is not typically necessary for walking with or without limb loss, this modification avoids biasing the optimization solution domain towards gaits with shallow flexion.

3. Torsional spring-dampers representing the passive stiffness of ligaments and other non-muscular joint structures were added to the joints (Anderson, 1999).

4. Eleven sphere-shaped Hunt-Crossley contact elements (Sherman, Seth & Delp, 2011) were added to the plantar surface of each foot (Fig. 1). The modulus (3.06 MPa) and damping coefficient (2.0 s/m) of the contact elements were set so that the heel contact sphere deformation and energy return were similar to the heel region of a human foot in an athletic shoe (Aerts & De Clercq, 1993). The frictional contact forces were a Stribeck function (Sherman, Seth & Delp, 2011) with respective static, dynamic, and viscous friction coefficients of 0.8, 0.8, and 0.5, and transition velocity 0.2 m/s.

The following additional modifications were not strictly necessary for Moco but were made for model fidelity:

1. Eight muscles actuating the lumbar joint (bilateral pairs of erector spinae, external obliques, internal obliques, rectus abdominus) and 12 muscles actuating each arm (biceps, brachialis, long and short heads of the triceps, anterior, middle, and posterior deltoid, pectoralis, latissimus dorsi, supraspinatus, infraspinatus, subscapularis) were added. The muscle geometry and parameters were taken from several sources (Anderson & Pandy, 1999; Holzbaur, Murray & Delp, 2005; Raabe & Chaudhari, 2016).

2. Muscle-specific time constants for activation and deactivation were defined as functions of mass and fast-twitch fiber fraction (Winters & Stark, 1985). Muscle mass was the product of physiological cross-sectional area, optimal fiber length, and density (Umberger, Gerritsen & Martin, 2003). Fast-twitch fiber fractions were referenced from Miller (2018).

3. The specific tension, which determines the muscle maximum isometric forces, was reduced to 40 N/cm2 from its original value of 60 N/cm2. With the original value, the model produced unrealistically high maximum isometric ankle torques.

4. Eight smaller muscles included in the popular “gait2392” OpenSim model but not in the original Rajagopal et al. (2016) model were added: bilateral pairs of gemelli, pectineus, peroneus tertius, and quadratus femoris. Extra small muscles help account for the fact that there were no reserve or residual actuators in the model, the model was actuated by muscles only.

Muscle metabolic rates were calculated using a smoothed version of the Bhargava, Pandy & Anderson (2004) energy model included in Moco. The non-muscular basal metabolic rate was calculated by assuming a default whole-body basal rate of 1.2 W per kg of whole-body mass, then subtracting the minimum muscle metabolic rates of 1.0 W per kg muscle mass. The values of 1.2 W/kg and 1.0 W/kg are approximate resting metabolic rates for the whole body of healthy young adult men (McMurray et al., 2014) and for human skeletal muscle (Elia, 1992), respectively. The whole-body metabolic rate was then the sum of the muscle metabolic rates and the non-muscular basal rate.

Post-limb loss model description

To model transfemoral limb loss, the 12 muscles spanning the right ankle joint were removed, and the muscles spanning the right knee joint had their attachment point moved proximally to locations eight cm from the distal end of the femur. The muscle reattachments and corresponding shortening of tendon slack lengths emulated a typical myodesis surgery as described by Ranz et al. (2017), preserving as much of the original muscle tension in the neutral hip position as possible. The right knee, ankle, subtalar, and toe joints had linear torque–angle relationships representing the prosthetic joints: (1) τ=−kθ−θ0−bθ ˙+τlimθ

where τ is the torque at the prosthetic joint, θ is the joint angle, k and b are prosthesis stiffness and damping parameters, and θ0 is the neutral prosthetic joint position. The right patella was removed, the mass of the right femur was reduced by 20%, and the mass of the right shank and foot were reduced by 50%. The resulting total mass of the transfemoral prosthesis model was 3.5 kg. Jaegers et al. (1993) reported a range of 2.9−4.0 kg for devices used by 11 individuals with transfemoral limb loss. Moments of inertia of the prosthetic bodies were reduced proportionate to these mass reductions. The moments of inertia of the prosthetic femur were further reduced by 30% to reflect that fact that mass is typically concentrated more distally in a prosthetic limb vs. a biological limb, easing the resistance to rotation. The function τlimθ provided a knee flexion torque that increased exponentially beyond full extension of the prosthetic knee, preventing hyperextension.

Equation (1) is a simple but reasonably realistic phenomenological model of many unpowered prosthetic ankles/feet, and of “mechanically passive” prosthetic knees that have a constant joint stiffness. However, many prosthetic knees have mechanisms that effectively alter the stiffness of the prosthetic joint, e.g., switching between a high-stiffness mode during the stance phase of walking and a low-stiffness mode during the swing phase. Examples include popular microprocessor-controlled prosthetic knees (Kaufman et al., 2008). OpenSim includes a “ClutchedPathSpring” actuator that switches between zero and non-zero stiffnesses based on a time-varying control signal. However, we had difficulty getting optimizations in Moco to converge when using a ClutchedPathSpring actuator. We instead used a simpler approach to model prosthetic knee phenomena more complex than Eq. (1) by adding an “ActivationCoordinateActuator” to the passive prosthetic knee model of Eq. (1): (2) τ=−kθ−θ0−bθ ˙+τlimθ+ατmax

where α∈−1,1 is the activation level and τmax = 100 Nm is the maximum active prosthetic knee torque. This actuator applied a generalized force, in this case the moment of force ατmax about the prosthetic knee flexion axis, in response to an input time-varying control signal that was transformed to the activation level α with first-order activation dynamics. The activation dynamics time constant was 100 ms. The prosthetic foot and ankle joints were still unpowered when using Eq. (2) at the knee. Equation (2) was not intended to simulate any specific make or model of prosthetic knee, but rather to check if the conclusions of the study depended critically on the assumption of a purely passive prosthetic knee represented by Eq. (1).

The post-limb loss model was otherwise identical to the pre-limb loss model. For example, the maximum isometric forces of the 108 muscles of the post-limb loss model were unadjusted from their values in the pre-limb loss model. No degrees of freedom were modeled at the interface between the prosthesis and the residual limb. The post-limb loss model was therefore more similar to an osseointegrated prosthesis implanted rigidly within the bone of the residual limb, rather than a traditional socket-based prosthesis.

Simulations

Virtual subjects

The computational speed of modern optimal control methods allows for simulations of multiple subjects, producing datasets that can undergo formal statistical analysis and hypothesis testing similar to studies on human participants. As a computer simulation study, the subjects in the present study were instances of the model with different model parameter values. The pre-limb loss model had a standing height of 1.70 m and body mass of 75.3 kg. Post-limb loss, the total body mass was 70.8 kg, with 67.3 kg of biological mass and 3.5 kg of prosthesis mass. This instance of the model was defined as “Subject 00”. Additional subjects were generated using the OpenSim “Scale Model” tool to scale the Subject 00 pre-limb loss model to the height and mass of each of the 14 able-bodied control participants from Russell Esposito, Ràbago & Wilken (2018), all of whom were United States military service members (mean age 26 years, range 22–44 years). The dimensions of all axes of all body segments in the model were scaled linearly by the ratio of the target height to the Subject 00 height. The mass of each body segment was scaled linearly by the ratio of the target total body mass to the Subject 00 total body mass. The segment dimension scaling also adjusted the muscle optimal fiber lengths and tendon slack lengths. Muscle maximum isometric forces were then scaled linearly by the ratio of the target whole-body mass to the Subject 00 whole-body mass.

Including Subject 00 and the 14 subjects scaled to the heights and masses from Russell Esposito, Ràbago & Wilken (2018), there was a total of 15 subjects in the study. We emphasize that these subjects were not intended to represent any specific individuals, and did not represent an exhaustive sensitivity analysis of the model’s parameter values. The element of subjects in the present study is better viewed as a limited sensitivity analysis, providing some confidence that any reported effects were due to the independent variable (limb loss) and generalize beyond the original model’s mass and height. The sample of 15 subjects had a minimum detectable effect size of 0.68 in the paired directional t-test of the change in metabolic cost pre- vs. post-limb loss, using standard long-term error rates of 5% for false-positives and 20% for false-negatives. In the present data, this effect size equated to a minimum detectable change in metabolic cost of about 1%.

Human participant

As a point of comparison to the simulations, a single high-functioning human participant with unilateral transfemoral limb loss completed an instrumented gait analysis of walking at their preferred “normal and comfortable” walking speed of 1.30 m/s. Ground reaction forces were measured at 1,000 Hz using piezeoelectric force platforms (Kistler). Positions of retroreflective markers on the body were measured at 200 Hz using 13 optical motion capture cameras (Vicon) and used to calculate joint and segment angles with standard six-degrees-of-freedom methods (Miller et al., 2014). The participant was fit with a chest worn heart-rate monitor (Garmin) and a wearable metabolic system which included a face mask and backpack-worn wireless transceiver (Cosmed K5). Metabolic rate was calculated from the measured steady-state pulmonary gas rates using Brockway’s equation (Brockway, 1987) with the urinary nitrogen term truncated. Participation was approved by the University of Maryland institutional review board, protocol #1826662. The participant gave written informed consent The participant was male, 26 years old, height 1.90 m, mass 96.3 kg, and had undergone limb loss due to traumatic injury. The participant was Medicare K-level 4, used a microprocessor-controlled prosthetic knee, and frequently participated in activities like weightlifting, running, and cycling.

Overview

The models were used to simulate periodic strides of walking at average speed 1.45 m/s and stride duration 1.0 s, values reported by Miller et al. (2014) for healthy adults walking at a subjective “normal and comfortable” pace. The simulations were posed as optimal control problems, finding the time-varying muscle excitations and associated model states that minimized the weighted sum of (i) deviations from average able-bodied gait mechanics and (ii) the gross metabolic cost, in the form of the cost function J: (3) J= ∫0T1NT∑i=1Nxit−μitσi2+w1mΔxE ˙b+E ˙mtdt

The first term on the righthand side of Eq. (3) is the “gait deviations tracking error”: xit is the value of model variable i at time t, μit is the mean of the analogous variable from the experimental gait analysis data of Miller et al. (2014), and σi is the standard deviation between-subjects of the experimental variable, averaged over the gait cycle. The N = 37 variables included in the tracking error term were the six translations and rotations of the pelvis, the 25 joint angles, and the three components of the ground reaction force under each foot. The square root of this term gives the average tracking error of the simulation, averaged over all timesteps of all variables, in multiples of the standard deviation. For example, a tracking error of 0.5 means that the simulation’s values for these 37 variables deviated from the average gait mechanics, defined by μit, by 0.5 standard deviations on average. For the motions of toe flexion, subtalar inversion, hip internal rotation, and the non-sagittal shoulder motions, we had low confidence in the accuracy of the experimental data, so we increased the value of σi to 3σi for these variables. This change in weighting allowed for better tracking of variables that can be measured with greater confidence in their accuracy, such as the sagittal joint motions and the ground reaction forces.

The second term on the righthand side of Eq. (3) is the metabolic cost: E ˙b is the non-muscular basal metabolic rate, E ˙mt is the sum of the muscle metabolic rates, m is the total body mass, and Δx is the horizontal displacement of the center of mass. The weighting constant w1 defines the relative emphasis the optimizer places on reducing the tracking error vs. the metabolic cost, which are generally in opposition (smaller tracking errors have greater metabolic costs). Values of w1 = 0.05 to 0.15 produced realistic metabolic costs of roughly 3–4 J/m/kg during model development with the Subject 00 pre-limb loss model, and a middle value of w1 = 0.10 was chosen. The same value (w1 = 0.10) was thereafter used in all simulations of all subjects, pre- and post-limb loss. We emphasize that the goal of these simulations was not to produce post-limb loss simulations that necessarily resemble the mechanics, energetics, and control of actual humans walking with a transfemoral prosthesis. Rather, the goal was to define specific clinically relevant objectives (walk with minimum deviations from a target gait and minimum metabolic cost) whose presence and relative weighting were identical for the pre- and post-limb loss simulations, such that differences between simulations could be confidently attributed to independent variable (limb loss) without a confounding effect of differences in the goals of the movement.

The optimal control problems were converted to nonlinear programming problems and solved using the direct collocation method in OpenSim Moco software (Dembia et al., 2020). The model’s state and control variables were discretized with a Hermite-Simpson transcription scheme, with 101 nodal values per variable spaced evenly over the stride duration, i.e., one node every 1% of the gait cycle. Simulation results in model development were invariant to finer grids tested up to 201 nodes per stride, at which point the computational speed of the simulations became impractically slow. A full stride was simulated because the assumption of bilateral symmetry involved in simulating a single step (Anderson & Pandy, 2001) is not valid for asymmetric limb loss locomotion. The nodal state and control variable values were optimized to minimize the value of Eq. (3), subject to constraints of the skeletal equations of motion, muscle activation and contractile dynamics, and task constraints of kinematic periodicity and average speed. The IPOPT solver within Moco was used for optimization (Wächter & Biegler, 2006), with constraint tolerance 1•10−4 and solver tolerance 1•10−3.

Validation simulations

Directly validating the model’s accuracy for predicting changes in metabolic cost after limb loss would require impractical longitudinal data on human metabolic cost pre- and post-limb loss. We instead gauged the model’s validity for predicting changes in metabolic cost with changes in lower limb mechanics that conceptually resemble limb loss. Previously, we demonstrated the model’s validity for predicting changes in metabolic cost with changes in foot mass and ankle stiffness (Miller & Russell Esposito, 2021). Here we added evaluations of the predicted changes in metabolic cost with changes in thigh mass and knee stiffness.

Browning et al. (2007) measured net metabolic rate of walking at 1.25 m/s with either +4 or +8 kg of mass added to each thigh (total of +8 or +16 kg). The data were well fit by a simple linear regression where the net metabolic rate increased by an average of 0.075 W/kg per kilogram of added mass. We added the same masses to the thigh segments of the Subject 00 pre-limb loss model and performed simulations of walking at 1.25 m/s using the same cost function and simulation approach described earlier. The model’s net metabolic rate in these simulations increased by 0.077 W/kg per kilogram of added mass.

Lewek, Osborn & Wutzke (2012) used a commercial knee brace that unilaterally limited the knee to about 30°  of flexion, and reported an average increase in net metabolic power of 17% when walking at 1.25 m/s compared to walking with unrestricted knee motion. We simulated knee bracing by adding a “CoordinateLimitForce” actuator to the right knee of the Subject 00 pre-limb loss model that made it essentially impossible for the model to flex its knee beyond 30°, and performed simulations of walking at 1.25 m/s using the same cost function and simulation approach described earlier. We were unable to achieve the magnitudes of metabolic rates reported by Lewek, Osborn & Wutzke (2012). For example, their baseline “control” condition with the knee brace unlocked had an average metabolic rate of 3.8 W/kg, which is unusually high for walking at 1.25 m/s. The model’s net metabolic rate at 1.25 m/s without the CoordinateLimitForce was 2.60 W/kg. The difference here was likely due to unmodeled aspects of the brace that affected the symmetry of the human participant’s walking gaits even with the brace unlocked. However, the model’s increase in metabolic rate with the CoordinateLimitForce on the right knee was +0.74 W/kg, which compares well to the increase in Lewek, Osborn & Wutzke (2012) of 0.66 ± 0.60 W/kg with the brace locked vs. unlocked.

Changing lower limb mass and joint stiffness of an intact biological limb is of course not equivalent to walking with a prosthetic lower limb. However, reasonably accurate model-based predictions of these changes provide confidence in using the model to predict changes in metabolic cost for conditions that cannot easily be validated directly with experiments.

The validity of the pre-limb loss model for predicting realistic muscle excitations when realistic motion and ground reaction forces in walking are imposed on it has been reported elsewhere (Lai, Arnold & Wakeling, 2017). As an additional gauge of the pre-limb loss model’s validity, Fig. 2 shows the on/off timing of excitations for 28 muscles in the baseline model’s pre-limb loss walking simulations, compared to normative timing measured by indwelling electromyograms from the same 28 muscles for walking in healthy adults (Sutherland, 2001). The post-limb loss excitation timings are also shown. Muscles in the model were deemed “on” when the excitation magnitude was above 3% on the range of 0–100%, and “off” otherwise. The main deviations in muscle on/off timing between the pre-limb loss simulation and the referenced electromyograms were the overall timing of the peroneal muscles and the lack of co-activation of all three biarticular hamstrings muscles, but otherwise the timing of the major muscles was reasonably similar to the referenced electromyogram timing.

Figure 2 Timing of simulated muscle excitations vs. able-bodied normative data.

Timing of when muscles in the model were “on” (defined as excitation above 3%) during the simulated stride for the pre-limb loss model’s right and left lower limbs, and the post-limb loss model’s prosthetic and intact limbs. ”Ref EMG” is data from indwelling needle electromyograms for healthy young adults (Sutherland, 2001). Muscles marked with an asterisk (*) were absent from the prosthetic limb. Muscles marked with two asterisks (**) had their geometry altered in the prosthetic limb.

It is difficult to make comparisons to electromyogram data from human participants with transfemoral limb loss since those datasets are sparse and rather variable across participants and across studies (e.g., Jaegers, Arendzen & de Jongh, 1996; Wetink et al., 2013), but the changes in excitation timing with limb loss were at least theoretically sensible, e.g., prolonged excitation of the prosthetic-side hip flexors. It is important to re-emphasize that the goal of these post-limb loss simulations was to change as little as possible about the model and the simulation workflow, other than replacing one biological limb with a prosthetic limb, such that the effect of the prosthetic limb could be isolated. We did not attempt to create models that include all typical differences between individuals with and without prosthetic limbs, nor to generate simulations that necessarily match the mechanics, energetics, and control of real humans who walk with a prosthetic limb. Other approaches are better suited to address those goals.

Pre-limb loss simulations

Simulations with the pre-limb loss models were performed using the direct collocation method. The same tracking targets μit and σi were used in the cost function (Eq. (3)) for all subjects. Each simulation was performed with three different initial guesses using simulation results from a previous similar study (Miller & Russell Esposito, 2021) and the result with the lowest cost function score was retained for analysis. When none of these initial guesses produced a result with a realistic metabolic cost, additional simulations with different initial guesses, specifically using the results of simulations from other “virtual subjects” described earlier as initial guesses, were performed until a result with a metabolic cost within two standard deviations of the mean from Das Gupta, Bobbert & Kistemaker (2019) was found.

Post-limb loss simulations

Simulations with the post-limb loss models were performed identically to the pre-limb loss simulations, with the exception that the standard deviations for the prosthetic limb’s knee, ankle, subtalar, and toe joints in Eq. (3) were increased to 10 σi, effectively reducing the weight on accurate tracking of these variables. The weight on these variables was reduced because the tracking targets μit were biological joint motions and it was expected that the model could not track these motions well with prosthetic joints. Motions of these joints in unpowered prostheses also typically do not closely resemble biological joint motions (e.g., Kobayashi et al., 2020). To approximate the patient-specific selection and adjustment process typically done in prosthesis prescription, the k, b, and θ0 prosthetic joint parameters in Eq. (1) were optimized along with the model states and controls in Moco for each subject’s post-limb loss simulation to minimize the cost function. Stiffness k was bounded on [1, 100] Nm/rad for the knee and [10, 1000] Nm/rad for the ankle. Damping b was bounded on [1, 10] Nm s/rad for both joints. Neutral joint position θ0 was bounded on [−10, 10] degrees for both joints. The prosthetic subtalar and toe joints were set to k = 400 Nm/rad, b = 1.0 Nm s/rad, and θ0 = 0°.

Post-limb loss simulations were performed twice for each subject, once with the passive prosthesis in Eq. (1) and once with the prosthesis that included the ActivationCoordinateActuator at the knee, Eq. (2). As noted earlier, the purpose of this actuator was to allow small deviations from the torque profile of the fully passive prosthetic knee, not to simulate a fully powered motorized prosthetic knee. This purpose was achieved by minimizing the control signal of the actuator in the cost function: (4) J= ∫0T1NT∑i=1Nxit−μitσi2+w1mΔxE ˙b+E ˙mt+w2u2tdt

where ut∈−1,1 is the prosthetic knee ActivationCoordinateActuator’s control signal. The weighting on this signal was given a high value of w2 = 100. Given the value of w1 = 0.1 in combination with w2 = 100, the cost of using the ActivationCoordinateActuator to produce the torque from the pre-limb loss simulation that tracked the target biological knee angle well equated to a very high increase in metabolic cost of +47 J/m/kg. The cost of using the powered prosthesis to alter the knee torque produced by the passive component of the prosthesis by an average of 1.0 Nm across the gait cycle equated to an increase in metabolic cost of +0.1 J/m/kg. The model with the ActivationCoordinateActuator (Eq. 4) will hereafter be referred to the “non-passive” prosthesis model. For initial guesses in the post-limb loss simulations, the pre-limb loss simulation result of the same subject was used first, then two other pre-limb loss simulations from subjects with heights closest to the present subject. If none of these guesses produced a result with a realistic metabolic cost, other initial guesses from other pre- and post-limb loss subjects were used until a result with a metabolic cost within two standard deviations of the mean from Das Gupta, Bobbert & Kistemaker (2019) was found.

Outcome variables and statistics

The main outcome variable of each simulation was the gross metabolic cost, defined as the metabolic energy expended per unit distance traveled by the model’s center of mass, divided by the body mass (units = J/m/kg). Paired Student’s t-tests were used to test the hypothesis that metabolic cost is greater post-limb loss vs. pre-limb loss, for both the passive and non-passive prosthesis models. The threshold for significance was p = 0.05, with Bonferroni adjustments for multiple comparisons.

Some studies on the limb loss population scale metabolic cost by the “total mass”, which includes the mass of the prosthesis (e.g., Jarvis et al., 2017). Others scale metabolic cost by the “biological mass”, excluding the mass of the prosthesis (e.g., Russell Esposito, Ràbago & Wilken, 2018). Because the prosthesis mass is a substantial fraction of the total body mass for individuals with transfemoral limb loss, the statistical analysis above was performed twice, once with the post-limb loss metabolic costs scaled only by biological mass, and a second time with the post-limb loss metabolic costs scaled by total mass. With two statistical comparisons to the pre-limb loss simulations with the different prosthesis models, and two comparisons with different mass scaling, the threshold for rejecting the null hypothesis was p = 0.05/4 = 0.0125.

Results

Gait mechanics

The pre-limb loss simulations deviated from the average able-bodied walking data of Miller et al. (2014) by 0.31 SD on average, with little variation in this tracking accuracy between subjects (range 0.30−0.32 SD). Average tracking errors in the post-limb loss simulations ranged from 0.41−0.47 SD and did not differ much between the passive and non-passive prosthesis simulations (0.43 ± 0.01 SD vs. 0.42 ± 0.01 SD). The power delivered about the prosthetic knee flexion axis by the ActivationCoordinateActuator in simulations with the non-passive prosthetic knee model was small, with a concentric peak of 0.6 ± 0.2 W in late stance and an eccentric peak of −0.5 ± 0.2 W in late swing (Fig. 3).

Figure 3 Knee moment and power of non-passive prosthetic knee model.

Moment and power delivered about the flexion axis of the prosthetic knee by the ActivationCoordinateActuator during walking simulations with the non-passive prosthetic knee model. The stride begins and ends at heel-strike of the right leg, which was the prosthetic leg. Lines are means for the 15 simulated subjects and shaded areas are one standard deviation around this mean.

Figures 4–7 show a typical tracking result, from the baseline model (Subject 00), for the pre- and post-limb loss simulations, using the passive prosthesis model in the post-limb loss case. Although the post-limb loss simulations did not attempt to track experimental gait data from human participants with limb loss, they still exhibited numerous prevalent gait deviations seen in human participants using transfemoral prostheses (Table 1). The magnitudes of these deviations were assessed in the high-functioning human participant with transfemoral limb loss and were generally similar to the post-limb loss simulations, with the exception that the human participant did not have a substantial deviation in their torso lateral bending range of motion (Table 1).

Figure 4 Ground reaction forces during the gait cycle.

Ground reaction forces (GRF) in the anterior, vertical, and medial directions during the stride cycle for the pre-limb loss (solid lines) and post-limb loss (broken lines) simulations. Data begin and end at heel-strike of the leg indicated in the legends heading each column. Shaded areas are one standard deviation around mean human experimental walking data (Miller et al., 2014).

Figure 5 Pelvis and lumbar kinematics during the gait cycle.

Pelvis and lumbar angles during the stride cycle for the pre-limb loss (solid lines) and post-limb loss (broken lines) simulations. Data begin and end at heel-strike of the right leg, which was the prosthetic leg post-limb loss. Shaded areas are one standard deviation around mean human experimental walking data (Miller et al., 2014).

Figure 6 Lower limb joint angles during the gait cycle.

Lower limb joint angles during the stride cycle for the pre-limb loss (solid lines) and post-limb loss (broken lines) simulations. Data begin and end at heel-strike of the leg indicated in the column headings. Shaded areas are one standard deviation around mean human experimental walking data (Miller et al., 2014). Panels marked with an asterisk (*) were variables with reduced tracking weight in the simulation cost function.

Figure 7 Upper limb joint angles during the gait cycle.

Upper limb joint angles during the stride cycle for the pre- limb loss (solid lines) and post-limb loss (broken lines) simulations. Data begin and end at heel-strike of the leg indicated in the column headings. Shaded areas are one standard deviation around mean human experimental walking data (Miller et al., 2014). Panels marked with an asterisk (*) were variables with reduced tracking weight in the simulation cost function.

Table 1 Gait deviations commonly seen in human participants with transfemoral limb loss that had magnitudes of at least 3° or at least 3% bodyweight (BW) in the post-limb loss simulations when using the passive prosthesis.

“Lesser” and “greater” refers to differences in comparison to the able-bodied pre-limb loss simulation. “Asymmetric” refers to the post-limb loss prosthetic limb vs. the intact limb. The “Prevalence” column is the fraction of 60 participants with unilateral transfemoral limb loss who exhibited these deviations in Carse et al. (2020). In the “Magnitude” columns, data labeled “Simulation” compare the post-limb loss vs. pre-limb loss simulations, or the post-limb loss prosthetic limb vs. intact limb, depending on the deviation in question. Data labeled “Human” compare the high-function human participant with transfemoral limb loss either to the referenced able-bodied human experimental means (Miller et al., 2014) or between their prosthetic vs. intact limb, depending on the deviation in question.

	Magnitude		
Deviation	Simulation	Human	Prevalence	
Lesser anterior ground reaction force peak on prosthetic side	4.9%BW	10.6%BW	100%	
Lesser posterior ground reaction force peak on prosthetic side	5.3%BW	4.1%BW	95%	
Asymmetric late-stance vertical ground reaction force peaks	12.1%BW	15.8%BW	81%	
Lesser early-stance knee flexion peak on prosthetic side	9.8°	11.7°	98%	
Lesser late-stance hip extension peak on prosthetic side	6.3°	7.6°	82%	
Greater average anterior pelvic tilt	3.1°	9.8°	77%	
Greater torso lateral bending range of motion	5.9°	0.6°	92%	

The post-limb loss simulations did not exhibit notable vaulting, hitching, or hip circumduction to swing the prosthetic limb, although these deviations were reported in only 20–54% of participants in Carse et al. (2020) and were also not observed in substantial magnitudes in the high-functioning human participant.

Metabolic cost

Gross metabolic cost in the pre-limb loss simulations was 3.22 ± 0.04 J/m/kg (mean ±SD). In human participants, average metabolic cost of walking at typical, comfortable speeds ranges from about 3.0−3.7 J/m/kg in young, unimpaired adults in most studies (Das Gupta, Bobbert & Kistemaker, 2019). The post-limb loss simulations with scaling by biological mass had metabolic costs of 3.52 ± 0.15 J/m/kg when using the passive prosthesis (Eq. 1)), and 3.42 ±  0.05 J/m/kg when using the non-passive prosthesis (Eq. 2). In comparison, the human participant with transfemoral limb loss and a microprocessor-controlled prosthesis had a metabolic cost of 3.56 J/m/kg when scaled by biological mass.

Metabolic costs for all simulations of all virtual subjects are shown in Fig. 8. Metabolic cost significantly increased in the post-limb loss simulations vs. the pre-limb loss simulations for all four cases tested, regardless of the prosthesis type or scaling method (all p <0.01). When scaling by total mass, metabolic cost increased by +0.7% (p = 0.0099) when using the non-passive prosthesis and by +3.9% (p = 0.0013) when using the passive prosthesis. When scaling by biological mass, metabolic cost increased by +6.0% (p = 3.4•10−12) when using the non-passive prosthesis and by +9.3% (p = 4.4•10−7) when using the passive prosthesis.

Figure 8 Metabolic cost of walking in the pre- and post-limb loss simulations.

Gross metabolic cost of walking in simulations with the pre-limb loss model, the post-limb loss model with a passive prosthetic knee, and the post-limb loss model with a non-passive prosthetic knee. Circles are individual subject results, lines connect the same subject between conditions, and squares are mean values in each condition. Data in (A) are scaled by biological body mass, i.e., excluding the prosthesis mass. Data in (B) are scaled by total body mass, i.e., including the prosthesis mass. Diamond symbol is data from a high-functioning young adult male with unilateral transfemoral limb loss using a microprocessor-controlled prosthetic knee.

Discussion

The purpose of this study was to determine if transfemoral limb loss increases metabolic cost of walking in optimal control simulations. The main finding was that converting one of the model’s limbs to a transfemoral prosthesis increased the metabolic cost of walking with minimum gait deviations, regardless of whether the prosthesis was passive or non-passive, and regardless of whether the metabolic cost was scaled by biological body mass or total body mass. Although this finding may seem intuitive given its alignment with experimental data (e.g., Gitter, Czerniecki & Weaver, 1995), the limb loss model had several advantages predisposing it to low metabolic costs: it had a long residual limb, rigid limb-prosthesis interface, and high level of muscle strength, and could optimize its prosthetic stiffnesses, all of which have been associated with low metabolic cost (Boonstra et al., 1994; Kooiman et al., 2023; Nolan, 2012; Fey, Klute & Neptune, 2012). The present results therefore suggest that unlike transtibial limb loss (Miller & Russell Esposito, 2021), transfemoral limb loss necessarily increases the metabolic cost of walking.

The main limitation of the present study is its modeling and simulation nature. While the post-limb loss simulations exhibited many of the gait deviations commonly seen in individuals with transfemoral limb loss (Table 1), we emphasize that the goal of these simulations was not to replicate reality in-silico or to emulate the gaits of any particular individuals. Rather, the goal was to isolate the effect of limb loss in and of itself on metabolic cost, which is not feasible in human participants. In experiments on human participants, testing of this effect is confounded by other variables that tend to be affected by limb loss and can independently affect the metabolic cost of walking, such as changes in walking speed or other stride parameters, changes in muscle strength, body composition, or other related variables like muscle quality, changes in other dimensions of fitness, and changes in sub-task priorities of walking. Computer modeling and simulation therefore allows for conclusions on the effect of transfemoral limb loss per se, independent of these confounders, within the limitations of modeling assumptions involved.

Relatedly, the increases in metabolic cost reported in these simulations of about 1–9% vs. the pre-limb loss case are much smaller than differences reported in experiments comparing human participants with and without transfemoral limb loss. This result is sensible given that, as noted earlier, the model had advantages predisposing it to low metabolic costs. In the most directly relevant comparison to the present simulations are studies on the military limb loss population, who compared to the general limb loss population are considered relatively young, strong, and fit, and often have access to more extensive rehabilitation and prosthetics prescription programs with higher-functioning goals such as return-to-duty. Metabolic cost in relatively young military service members with transfemoral limb loss was on average 45% greater than age-matched able-bodied service members when scaled by biological body mass (Russell Esposito, Ràbago & Wilken, 2018), and 20% greater when scaled by total body mass (Jarvis et al., 2017). We speculate that this difference between simulations and human experiments is because the assumption in the simulations of maintenance of muscle strength between the pre- and post-limb loss conditions is difficult to achieve in reality for transfemoral limb loss. An optimistic interpretation of this difference between the present simulations and previous experiments is that while transfemoral limb loss may invariably increase the metabolic cost of walking, this increase does not necessarily need to be large and may be reduced by modifiable factors of strength and fitness. Nolan (2012) reported that a 10-week program of 20 strength-training sessions of the prosthetic-side hip muscles reduced the oxygen consumption rate of walking at 1.0 m/s by an average of 7%, and enabled treadmill running when the participants with transfemoral limb loss all reported inability to run pre-training.

A related point of comparison on what may be achievable through modifiable factors post-limb loss is the single human participant in the present study. This individual was included because they are unusually young, fit, and mobile even compared to other high-functioning individuals with transfemoral limb loss. As an example of this individual’s very high level of function, they were able to hop vertically on their prosthetic limb, which requires remarkable levels of prosthetic-side strength, balance, and confidence. This individual’s metabolic cost of walking was only slightly above the range of costs in the post-limb loss simulations (Fig. 8), suggesting that the simulation result reported here of a small increase in metabolic cost post-limb loss may also be achievable in human participants. However, this suggestion is cautioned by the acknowledgement that this individual’s pre-limb loss metabolic cost is unknown.

As noted earlier, the post-limb loss simulations did not explicitly attempt to track gait mechanics data from individuals with limb loss, but nonetheless exhibited many of the common gait deviations associated with transfemoral limb loss (Table 1). This result suggests that these deviations may be unavoidable, at least qualitatively, in walking with transfemoral limb loss, when combined with the goal of a relatively low metabolic cost. Attempting to further suppress these deviations, i.e., track the able-bodied experimental data more closely, increases the simulated metabolic cost. Previous simulations and experiments on human participants have also suggested that altering post-limb loss gait mechanics to be more symmetric increases metabolic cost (Handford & Srinivasan, 2016; Wedge, Sup & Umberger, 2022).

The simulations with the non-passive prosthesis model (Eq. 2) had lower metabolic cost than simulations with the passive prosthesis model (Eq. (1), by an average of −2.9% regardless of the method of body mass scaling (Fig. 8). We emphasize that the non-passive prosthesis model was not intended to emulate powered, motorized, robotic, etc. prosthetic knees. Rather, the high weight placed on the prosthetic knee’s excitation control (parameter w2 in Eq 4) allowed the knee to deviate only slightly from a fully passive joint moment profile without increasing metabolic cost or gait deviations, and was included among the conditions tested here to determine of the results depended critically on the passive prosthetic knee model assumed in Eq. (1). Popular battery-powered, microprocessor-controlled knees generally function differently than Eqs. (2) and (4) in that they modulate the level of damping in the joint based on feedback signals, but they achieve a conceptually similar effect to Eqs. (2) and (4) in that a small amount of power is used to deviate slightly from a fully passive knee moment profile. Kaufman et al. (2008) reported an average decrease in oxygen cost of walking of −2.3% when using microprocessor-controlled vs. passive mechanical prosthetic knees after an average of 18 weeks of acclimation, although this difference was not statistically significant. Other conceptually similar studies have reported similar results (Wong et al., 2012). Powered prostheses that more closely emulate able-bodied knee mechanics, or that restore ground contact sensation, have achieved larger reductions in metabolic cost of walking in case studies (Martinez-Villalpando et al., 2011; Petrini et al., 2019). Since the post-limb loss simulations were unable to achieve the pre-limb loss metabolic costs, the present results suggest that powered prosthetic joints may be needed to achieve able-bodied metabolic costs of walking after transfemoral limb loss.

The choice of scaling by biological body mass vs. total body mass affected metabolic cost by roughly 5% (Fig. 8), which can feasibly affect conclusions since the minimum detectable difference using modern pulmonary gas measurements is roughly 2% (Davidson, Gardinier & Gates, 2016; Guidetti et al., 2018). For example, the +0.7% change in metabolic cost scaled by total mass with the non-passive prosthesis, while statistically significant, is not reliably detectable in human experiments even if it is deemed meaningful, while the change of +6.0% in this same condition when scaling by biological mass is easily detectable. We suggest that scaling by biological body mass casts metabolic cost as an index of efficiency (energy expended per unit mass consuming that energy), while scaling by total body mass is an index of economy (energy expended per unit mass transported). The option of no scaling is appropriate when absolute energy expenditure is of interest. Regardless, the scaling choice should be clearly reported and the sensitivity of conclusions to this choice should be considered.

As noted in the methods, the prosthesis model in the present simulations assumed a rigid attachment between the prosthesis and the residual limb, more similar to a bone-anchored osseointegrated prosthesis than a traditional socket-based prosthesis. The residual transfemoral limb moves within the socket during walking by rotations of 7−11°  and displacements of 1–3 cm, and some of these motions correlate with prosthesis- and health-related problems (Gale et al., 2020). Walking with an osseointegrated prosthesis generally has a lower metabolic cost than walking with a socket-based prosthesis (Van de Meent, Hopman & Frölke, 2013; Kooiman et al., 2023), although it is unclear if this difference is due to the rigidity of the limb-prosthesis interface specifically. In model development, adding a degree of freedom that allowed for pistoning between the prosthesis and residual femur increased metabolic cost by 1%. We did not attempt more complex multi-dimensional limb-prosthesis interface models, but comparing these models to the rigid osseointegration-type model would help to isolate the effect of osseointegration per se, independent from potential confounders such as baseline mobility and rehabilitation, which will be an important question as osseointegration becomes a more popular treatment option for individuals with limb loss.

A final point of a technical nature may also have relevance to the challenge of learning to walk with a transfemoral prosthesis. Compared to our previous study that used a similar model to simulate walking with a transtibial prosthesis (Miller & Russell Esposito, 2021), the present simulations with a transfemoral prosthesis were much more difficult optimization problems for the Moco/IPOPT software, requiring on average twice as many different initial guesses than the previous study to converge on a high-quality simulation result for each subject: good tracking similar to Figs. 4–7, realistically low metabolic costs seen in Fig. 8. The simulation results from other initial guesses not shown here tended to have very poor tracking of the target experimental data and/or unrealistically high metabolic costs on the order of 10 J/m/kg. One subject did not converge on any result close to their result in Fig. 8 until the 14th initial guess. Researchers attempting similar simulations with similar methods should be aware that an arbitrary initial guess may not produce good results, many initial guesses may need to be tried.

Some additional limitations are worth highlighting to aid readers in gauging their confidence in the present results before concluding. First, with the high-dimensional solution domain and numerical method used in the optimizations, we cannot guarantee that all the simulations are near their global minimum in their solution domain. It is possible that post-limb loss simulations with lower metabolic costs than those shown in Fig. 8 exist, which could feasibly change conclusions reached. We performed all simulations of all subjects with at least three different initial guesses and in some cases as many as 14 initial guesses in attempting the minimize the impact of this limitation. We also checked on the sensitivity of the results to the five results in Fig. 8 that were above the mean for the post-limb loss passive prosthesis condition, and the statistical outcome of a significantly greater metabolic cost post-limb loss was robust to their exclusion. Second, while we presented data gauging the model’s ability to predict small changes in metabolic cost, its ability to predict pre- vs. post-limb loss changes in metabolic cost has not been validated, as there are no data in the literature to base this validation on. There are longitudinal data on changes in metabolic cost with osseointegrated vs. socket-based transfemoral prostheses (Van de Meent, Hopman & Frölke, 2013), which would be a useful addition to the present model’s validity. Third, to our knowledge this is the first study to perform these types of simulations with a model of a prosthetic knee. Options for modeling a prosthetic knee in the current OpenSim Moco software were limited, and the prosthetic knee models used here (Eqs. (1) and (2) were rather simple. Prosthetic knee design characteristics have had small and inconsistent effects on metabolic cost (Wong et al., 2012; Sawers & Hafner, 2013), so we do not think the simplicity of the knees modeled here is a major limitation. For example, preliminary tests in model development of polycentric vs. single-axis knee rotation centers did not affect metabolic cost of the post-limb loss simulation.

Conclusions

Replacing a biological limb of a three-dimensional full-body musculoskeletal model with a transfemoral prosthetic limb increase the metabolic cost of walking with minimal gait deviations, and induced many of the gait deviations commonly seen in human participants walking with transfemoral prostheses, even without any post-limb loss changes in muscle strength. However, the magnitude of the increase in metabolic cost was considerably smaller than differences in metabolic cost typically seen in human participants with transfemoral limb loss vs. able-bodied participants, particularly when a small amount of power (peak ∼0.5 W) was delivered to the prosthetic knee. These results suggest that a large increase in metabolic cost post-transfemoral limb loss may be avoidable in the long-term by maintenance of muscle strength and prosthetic knee technology.

Additional Information and Declarations

Competing Interests

Author Contributions

Human Ethics

Data Availability

Ross H. Miller is an Academic Editor for PeerJ.

Ross H. Miller conceived and designed the experiments, performed the experiments, analyzed the data, prepared figures and/or tables, authored or reviewed drafts of the article, and approved the final draft.

Elizabeth M. Bell performed the experiments, analyzed the data, authored or reviewed drafts of the article, and approved the final draft.

Elizabeth Russell Esposito conceived and designed the experiments, authored or reviewed drafts of the article, and approved the final draft.

The following information was supplied relating to ethical approvals (i.e., approving body and any reference numbers):

University of Maryland

The following information was supplied regarding data availability:

All OpenSim models and simulation data are available at SimTK: https://simtk.org/projects/umocod.

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
