# Peer review of "Transfemoral limb loss modestly increases the metabolic cost of optimal control simulations of walking"

_PeerJ, doi:10.7717/peerj.16756_

## Round 0.1 · original submission · Major Revisions

Two reviewers have given enthusiastic and constructive reviews for this paper, and I concur that it's excellent science and well written. There are moderate revisions needed, and some re-review to check that they are sufficient. Points include better justification of muscle parameters and clarification of limb center of mass, plus careful consideration of cost and objective functions. We look forward to seeing the revised manuscript.

Reviewer 1 ·

Basic reporting

No comment.

Experimental design

No comment.

Validity of the findings

No comment.

Additional comments

Overall, the study provided is well-constructed and very clearly presented. The authors have taken extra care to describe all aspects of the simulation methodology and how they are relevant to the study's hypothesis. The table and figures are clear and directly address the goals of the study. I only have a handful of minor comments that should be addressed before acceptance.

Minor comments:

Line 161: The authors provide a time constant for the “ActivationCoordinateActuator”, but should also provide a brief description of the model itself (e.g., generalized force with a first-order activation dynamics model, etc).

Line 342: The authors mention a “target biological knee torque waveform”, but I don’t believe there is a knee torque tracking term in the cost function. Perhaps this should be rephrased to “the knee torque required to track the biological knee joint angle”.

Lines 392-393: Based on Fig 7, the metabolic cost values reported here seemed to be swapped (i.e., the passive prosthesis solution should have a higher metabolic cost compared to the non-passive prosthesis solution).

Line 400: A word is missing in this sentence: “Metabolic cost increased significantly…”.

·

Basic reporting

Overall, this paper is extremely well written that was very enjoyable to read. I have no comments on this area, aside from potentially including more detail/data on the initial guesses with the raw data that was supplied.

Experimental design

L116 – L119: Muscle physiological parameters (activation/deactivation time constants, muscle mass, maximum isometric strength) were based on data in the literature from healthy individuals, which is an understandable decision given the paucity of data in this realm. The authors appropriately acknowledge the lack of inclusion of strength data as a limitation that may influence metabolic cost as it is known to be altered by transfemoral amputation. However, what about the activation parameters? It is possible that these parameters are also influenced by amputation (particularly due to increased fatty infiltration that influences contraction dynamics), and thus potentially metabolic cost.

L146 – L150: Was there no change to the position of the center of mass in the post-limb loss model? As amputation will result in a more proximal center of mass position (Ferris et al., 2017),which in turn will affect the lever arm relative to the hip joint, it is likely that this will influence metabolic cost most prominently during the swing period. Have the authors considered this in their model? If not, a justification as to why not would be beneficial given the level of detail included in other aspects of the model.

L252 – L254: This sentence is not quite clear. If part of the goal of the objective function was to minimize error between experimental and computational degrees of freedom, how could the goal not be to reproduce data from actual humans? Please clarify.

L273 – L280: The current validation approach is to evaluate if changes in model mass and knee stiffness appropriately resulted in changes in metabolic cost. While this is important to determine, this does not necessarily serve as model validation as an invalidated pre limb loss model could still produce a systemic change in metabolic cost. While I acknowledge the author’s statement that validating would require longitudinal data across amputation, which does not exist, there are other validation parameters that could be considered. For example, what do the pre and post limb loss muscle activation parameters compare to EMG recordings within the literature? What are the model reserves?

Ferris, A.E., Smith, J.D., Heise, G.D., Hinrichs, R.N., Martin, P.E., 2017. A general model for estimating lower extremity inertial properties of individuals with transtibial amputation. J Biomech 54, 44-48.

Validity of the findings

As described in my comments in the Experimental Design and General Comments section of the review, two additional areas are needed/would be helpful surrounding the validity of the findings: 1) revised validation and 2) more details surrounding the initial guesses.

Additional comments

L311 – L313 states that 3 initial guesses from a prior TTA model were used for the pre-limb loss models, yet it is not entirely clear as to what was used for the post-limb loss model. L525 – L528 indicates that up to 14 initial guesses were used for one model with a TFA. What changed between guesses? How were these decisions made? The authors describe how important this is in L528 – L530, thus providing more details surrounding the initial guesses, with potential inclusion in the provided data, would be beneficial from a replication standpoint.

L325 – L329: Were the prosthesis parameters actually optimized within the objective function to minimize the cost function? Or were these parameters manually tuned to produce an adequate solution? More clarification, including potentially rephrasing the term ‘optimized’ on L327, would be helpful.

L326: ‘typical’ should be replaced with ‘typically’

L400 – L404: This sentence could be broken up as it is very hard to digest. Additionally, a word may be missing “metabolic cost significantly post-limb loss vs. pre-limb loss.”

---

## Round 0.2 · accepted · Accept

Sorry for the delay in returning this decision, but we have checked the revised MS and are satisfied with the changes, which have markedly improved the study. Congratulations on the acceptance of your paper!

·

Basic reporting

The authors have done a thorough job on addressing all reviewers comments. I commend them on their work. The paper is improved, and represents an important contribution to the literature.

Experimental design

Thank you for addressing the individual points. The clarity provided helps the interpretation of the paper.

Validity of the findings

The inclusion of the model activations to experimental data is helpful, as are the rest of the changes to the validation methodology.